# Identifying Chemical Composition, Safety and Bioactivity of Thai Rice Grass Extract Drink in Cells and Animals

**DOI:** 10.3390/molecules26226887

**Published:** 2021-11-15

**Authors:** Suthaya Phimphilai, Pimpisid Koonyosying, Nuntouchaporn Hutachok, Tanyaluk Kampoun, Rufus Daw, Chaiyavat Chaiyasut, Vanli Prasartthong-osoth, Somdet Srichairatanakool

**Affiliations:** 1Division of Science and Food Technology, Faculty of Engineering and Agro-Industry, Maejo University, Chiang Mai 50290, Thailand; sphimphi@gmail.com; 2Oxidative Stress Research Cluster, Department of Biochemistry, Faculty of Medicine, Chiang Mai University, Chiang Mai 50200, Thailand; pimpisid_m@hotmail.com (P.K.); thenuntouch@gmail.com (N.H.); tanyalukkk@hotmail.com (T.K.); rufus.daw@postgrad.manchester.ac.uk (R.D.); 3School of Biological Sciences, Faculty of Biology, Medicine and Health, University of Manchester, Manchester M13 9PL, UK; 4Department of Pharmaceutical Sciences, Faculty of Pharmacy, Chiang Mai University, Chiang Mai 50200, Thailand; chaiyavat@gmail.com; 5Natural Rice Company Limited, Sawankalok, Sukhothai 64110, Thailand; sukho.organicrice@gmail.com

**Keywords:** antioxidant, catechins, iron-chelating, *Oryza sativa*, rice, toxicity

## Abstract

Rice grass has been reported to contain bioactive compounds that possess antioxidant and free-radical scavenging activities. We aimed to assess rice grass extract (RGE) drink by determining catechin content, free-radical scavenging and iron-binding properties, as well as toxicity in cells and animals. Young rice grass (Sukhothai-1 strain) was dried, extracted with hot water and lyophilized in a vacuum chamber. The resulting extract was reconstituted with deionized water (260 mg/40 mL) and served as Sukhothai-1 rice grass extract drink (ST1-RGE). HPLC results revealed at least eight phenolic compounds, for which the major catechins were catechin, epicatechin and epigallocatechin-3-gallate (EGCG) (2.71–3.57, 0.98–1.85 and 25.47–27.55 mg/40 mL serving, respectively). Elements (As, Cu, Pb, Sn and Zn) and aflatoxin (B1, B2, G1 and G2) contents did not exceed the relevant limits when compared with WHO guideline values. Importantly, ST1-RGE drink exerted radical-scavenging, iron-chelating and anti-lipid peroxidation properties in aqueous and biological environments in a concentration-dependent manner. The drink was not toxic to cells and animals. Thus, Sukhothai-1 rice grass product is an edible drink that is rich in catechins, particularly EGCG, and exhibited antioxidant, free radical scavenging and iron-binding/chelating properties. The product represents a functional drink that is capable of alleviating conditions of oxidative stress and iron overload.

## 1. Introduction

Known endogenous antioxidant sources are superoxide dismutase (SOD), catalase (CAT) and glutathione peroxidase (GPx), glutathione reductase, glutathione (GSH) and their reducing equivalents, while exogenous sources include vitamin C, vitamin E, vitamin A, β-carotene, selenium and dietary polyphenols [1]. Antioxidants play an important role in reducing the numbers of net reactive oxygen species (ROS) under oxidative stress, ensuring cell viability and implementing certain beneficial health measures. Most investigatory applications of antioxidants focus on lipophilic molecules and functional foods [2]. Accordingly, regular ingestion of dietary polyphenols may reduce the risk of certain cancers, possibly by interference with cancer cell proliferation and stimulation of host immune systems [3]. Certain polyphenols, including catechin (C), epicatechin (EC), epigallocatechin (EGC), epicatechin-3-gallate (ECG) and epigallocatechin-3-gallate (EGCG), are found in the form of secondary metabolites in numerous plants [4]. They are known to exert anti-oxidative and free-radicals scavenging properties due to the aromatic hydroxyl groups present at the two phenolic rings [5]. In particular, galloylatedcatechins exhibit higher anti-oxidative properties in comparison to their nongalloylated versions [6]. Moreover, C, EC, ECG and EGCG have shown an ability to inhibit plasma lipid peroxidation in tissue samples, microsomes and plasma low-density lipoproteins (LDL) [7]. Additionally, green tea-derived EGCG and ECG exhibited iron-chelating properties by lowering levels of iron overload and oxidative stress in β-thalassemic mice [8]. These molecules could pass through the cell membrane with a significant degree of efficacy by acting as a powerful antioxidant and chelator source [9].

Whole rice (*Oryza sativa* L. family Gramineae) grains vary in terms of the amounts of starch, γ-oryzanol, essential amino acids, proanthocyanidins, anthocyanins, γ-aminobutyric acid and iron compounds they contain [10]. Importantly, rice grass (or rice seedling) (RG) drinks have been biotechnologically produced to improve certain nutritional qualities and health benefits; however, very few investigatory research studies have been conducted [11]. Recently, RG drinks have been found to possess antioxidant and free-radical scavenging activities and have been proposed as a functional food [12,13]. Herein, we hypothesize that rice grass would be abundant with catechin derivatives in a similar manner to green tea and could be beneficial for their biological functions in living organisms. In this study, we have focused on analyzing the active compounds of Sukhothai-1 rice grass (ST1-RG) drinks in order to determine their antioxidant ability and to study their possible toxicity in cells and animals.

## 2. Results

### 2.1. Catechin Contents

Using high-performance liquid chromatography-diode array detection (HPLC-DAD) analysis, ST1-RG (lot 1), ST1-RG (lot 2) and ST1-RG (lot 3) extracts were used to prepare drinks containing C, EC and EGCG with retention times (T_R_) of 10.4, 12.4 and 20.7 min, respectively (Figure 1). Thus, the average amounts of C, EGCG and EC in the three different ST1-RG extracts (ST1-RGE lots 1–3) were 11.9, 102.3 and 5.1 mg/g, respectively. Accordingly, the EC contents in the ST1-RGE lots 2 and 3 were significantly lower than that in the ST1-RGE lot 1 and they comprised 3.10, 26.6 and 1.32 mg/40 mL servings, respectively (Table 1). Apparently, EGCG was the most abundant catechin, and C was the most second abundant catechin; however, the amounts of the two catechins were not determined to be significantly different among the three different ST1-RG samples.

### 2.2. Microbiological Test

The total plate count was recorded at 6.7 × 10^3^–7.1 × 10^3^ CFU/mL, for which pathogenic bacteria, including *Escherichia coli* and *Staphylococcus aureus*, were found to be less than 1.1 of the minimal possible number (MPN)/mL and less than 1.0 of the colony-forming unit (CFU)/mL. Moreover, the total amount of aflatoxins in the ST1-RGE powder was found to be less than 0.25 μg/kg (part per million, ppm), for which the amounts of aflatoxin B1, B2, G1 and G2 were less than 0.25 μg/kg (ppm).

### 2.3. Stoichiometric Analysis of Metals

Inductively coupled plasma mass spectrometry (ICP-MS) analysis revealed the presence of the following contents in 1 kg of ST1-RGE: 4.03 mg of arsenic (As), 17.46 mg of copper (Cu), 0.63 mg of lead (Pb), 4.48 mg of tin (Sn) and 68.92 mg of zinc (Zn). These amounts were equivalent to 1.05, 4.53, 0.16, 1.16 and 17.92 μg/40 mL serving, respectively, while mercury (Hg) was not detectable at all (Table 2). Importantly, all of the amounts did not exceed the relevant limits when compared with WHO guideline values as follows: As (10 mg/kg), Cu (2000 mg/kg), Pb (10 mg/kg), Sn (250 mg/kg), Zn (3.0 mg/kg) and Hg (0.0038 mg/kg) [14]. Furthermore, a study on product shelf life revealed that the hot-air oven- and vacuum-processed ST1-RG samples were stable for at least six months when kept in polypropylene bags under ambient temperatures.

### 2.4. Free-Radical Scavenging Activity

#### 2.4.1. DPPH Radical Scavenging Activity

We found that 2,2-diphenyl-1-picrylhydrazyl radical (DPPH^•^) generation was inhibited by ST1-RGE and Trolox treatments in a dose-dependent manner with concentrations at 50% inhibition (IC_50_) values of 0.15 mg/mL and 0.01 mg/mL, respectively (Figure 2).

#### 2.4.2. Hepatic ROS Scavenging and Anti-Lipid Peroxidation Activities

Interestingly, the ST1-RGE treatment was found to reduce reactive oxygen species (ROS) levels effectively in H_2_O_2_-induced human hepatocellular carcinoma (HuH7) cells in a concentration-dependent manner (Figure 3). Accordingly, levels of thiobarbituric acid-reactive substance (TBARS) were significantly increased in HuH7 cells that had been loaded with 1 mM ferric ammonium citrate (FAC), while hepatic TBARS levels were dramatically decreased by DFP (9 μg/mL) (*p* < 0.05) and ST1-RGE (3.3–40 μg EGCG equivalent/mL) (Figure 4). Taken together, ST1-RGE that was abundant with phenolic catechins, particularly EGCG, could exert powerful radical-scavenging and anti-lipid peroxidation activities, not only in the aqueous environment but also in the cells.

#### 2.4.3. Serum SOD Activity in Rats

Interestingly, ST1-RGE treatments at 50 and 100 mg/kg BW tended to slightly decrease serum levels of superoxide dismutase (SOD) activities, for which 50 mg ST1-RGE/kg BW was better than 100 mg ST1-RGE/kg BW when compared with the DI treatment (Figure 5).

### 2.5. Iron-Chelating Activity

Similar to desferrioxamine (DFO) and DFP actions, ST1-RGE was found to bind iron, as well as both ferrous ion (Fe^2+^) and ferric ion (Fe^3+^). ST1-RGE was also found to be capable of existing in an aqueous environment of pH 7.0 in a concentration-dependent manner until saturation was reached (Figure 6a,b). Moreover, ST1-RGE was able to chelate redox-active non-ntransferrin bound iron (NTBI) in thalassemic serum effectively, but not completely, in a concentration-dependent manner. However, ST1-RGE chelation was less effective than DFO and DFP (Figure 6c).

### 2.6. Toxicity

#### 2.6.1. Toxicity in Hepatocyte Culture

ST1-RGE treatments (0.31–20 μg/mL) were not found to be toxic (cell viability > 80%) to HepG2 cells for up to 24 h (Figure 7). Similarly, the ST1-RGE treatments (0.31–20 μg/mL) were not found to be toxic to the Huh7 cells for up to 48 h; however, toxicity was observed at 20 μg/mL-ST1-RG of the drink treatment for 48 h but nonsignificantly (*p* = 0.24) when compared without treatment.

#### 2.6.2. Acute Toxicity in Mice

All study mice were alive and did not exhibit any clinical signs or symptoms throughout the 14 days of the study. Furthermore, their autopsied internal organs did not reveal any pathological appearance. The findings demonstrate that the median lethal dose (LD_50_) value of the ST1-RGE was higher than 5.0 g/kg.

#### 2.6.3. Sub-Chronic Toxicity in Rats

The results indicate that the body weight (BW) of the rats treated with DI and ST1-RGE (50 and 100 mg/kg BW) increased in accordance with time, whereas BW changes were not significantly different when compared with DI (Figure 8).

Although white blood cell (WBC) numbers for rats treated with 50 mg and 100 mg of ST1-RGE tended to increase in both regular and satellite studies when compared with the DI group, these levels were still within a normal range. In addition, percentages of neutrophil, lymphocyte, monocytes, eosinophil and basophil in the blood of treated rats were not significantly different from those of the control rats. Similarly, levels of red blood cells (RBC) and platelet (PLT) indices fell within normal ranges and were not changed by treatment with the ST1-RGE (Table 3a–c). The findings imply that consumption of ST1-RGE did not influence the normal functions of bone marrow cells in rats.

As shown in Table 4, levels of blood urea nitrogen (BUN) and serum creatinine (CRE) for all groups fell to normal ranges, while the BUN level was not found to be different but the CRE level was significantly increased after treatments with ST1-RGE (50 and 100 mg EGCG equivalent/kg BW). Level of serum albumin (Alb) was decreased significantly after the treatment with ST1-RGE (50 mg EGCG equivalent/kg BW). Accordingly, the levels were not observed to be significantly different before and after treatment, even in the satellite study. Serum levels of aspartate aminotransferase (AST) and alanine aminotransferase (ALT) activities on day 90 increased by approximately 1.5–2 folds in all study groups when compared with those at the beginning of the study; however, ST1-RGE treatments were found to increase levels of ALT and AST activities (*p* < 0.05) when compared with the DI treatment in both standard and satellite studies. Inversely, serum levels of alkaline phosphatase (ALP) activities on day 90 decreased for rats in all groups, for which the results of the ST1-RGE treatments did not affect ALP levels when compared with results of the DI treatment in both standard and satellite studies. Moreover, level of low-density lipoprotein cholesterol (LDLC) was increased (*p* < 0.05). Nonetheless, there were no clinically adverse findings of serum lipids, including triglyceride (TG), total cholesterol (TC) and high-density lipoprotein cholesterol (HDLC) associated with ST1-RGE administration (50 and 100 mg/kg BW), on days 0, 45 and 90 when compared with the DI group. The results imply that ST1-RGE treatments (50 and 100 mg/kg BW) of the rats for 90 d may have interfered kidney and liver functions or serum lipid profiles, apparently causing increases in levels of creatinine, ALT and AST activities.

At the end of the experiment, mice were sacrificed and their internal organs, including heart, liver, kidneys, adrenal glands, spleen, brain, lungs, testes and ovaries, were excised, weighed and examined pathologically. Organ wet weights are shown in Table 5. There were no significant changes in the organ weights of male and female rats, except kidneys and adrenal glands, in the DI group and the ST1-RGE treatment groups for both the standard study and the satellite study. Macroscopic examinations revealed no gross abnormalities related to the treatment with ST1-RGE. Moreover, there were no treatment-related histopathological findings (hematoxylin and eosin-stained tissues) in all treated mice at the end of study. Nevertheless, incidental findings, such as slight spotty necrosis cells and lymphocyte infiltration in the livers, were reported without obvious differences among subjects in the control group, while no differences were observed in doses of the ST1-RGE treatment after administration over 90 d. Hence, no outcomes were accompanied by histopathological changes that would suggest toxicological relevancy to the ST-RGE treatments at two doses.

## 3. Discussion

Lipophilic and hydrophilic antioxidants, such as phenolic acids, flavonoids and catechins, are commonly found in pigmented rice (*Oryza sativa* L.) bran and grains [15]. Under conditions of salt-induced ionic toxicity and osmotic stress, rice grass (seedlings) develop an ROS detoxifying system by increasing the contents of phenolic compounds and flavonoids, along with the activities of SOD, catalase and other reductases, in order to ameliorate oxidative stress [16]. In other studies, rice grass juice was found to contain various types of phenolic compounds, predominantly flavone glycosides, such as chrysoeriol arabinosyl arabinoside derivatives, that exert strong antioxidant properties [12,17]. Currently, HPLC-DAD results have demonstrated that the methanolic extracts of rice seedlings (strain Q1) contained phenolic compounds (770 ± 22 μg/g dry weight of the extract) identified as gallic acid, chlorogenic acid, caffeic acid, *p*-coumaric acid, rutin, ferulic acid, salicylic acid and kaempferol (80.4, 54.7, 217.5, 58.5, 193.8, 11.6, 20.2 and 33.3 μg/g of the extract) and exerted DPPH^•^ scavenging activity [18]. In the present study, C, EGCG and EC were detected in both of the hot water extracts of ST1-RG and ST1-RG drinks by HPLC through comparisons with their authentic standards. Among the detected compounds, EGCG was found in the highest amounts in both ST1-RG extracts and ST1-RG drinks, followed by C and EC, respectively. According to HPLC analysis, the contents of the detected compounds in both types of samples were not significantly different for the three separated extractions of rice grass. Hence, an effective method of quality control was applied to the extraction method used to prepare and preserve the rice grass extract drinks and related products. Moreover, we also tested for any instances of contamination of bacteria, aflatoxins and elements in the products. According to the provisional guidelines, levels of all contaminants in both types of samples were lower than the relevant limit values. These results suggest that the products containing C, EGCG and EC were safe for consumption.

Oxidative stress occurs when an imbalance exists between ROS production and antioxidant capacity, leading to the progression of oxidative stress-related diseases. Thus, antioxidants are an important key for improving oxidative stress and for preventing both noncommunicable diseases and metabolic syndromes. Herein, both Trolox and ST1-RGE scavenged DPPH^•^ in a concentration-dependent manner, for which Trolox exhibited greater scavenging activity than ST1-RGE. Likely, ST1-RGE contained various types of phenolic compounds that were capable of scavenging DPPH^•^ through their detected antioxidant compounds including C, EGCG and EC. Similarly, a previous study reported that C, EGCG and EC effectively scavenged DPPH^•^ through electron donation from the hydroxyl groups in their structures [19]. In addition to DPPH^•^ scavenging assay, we also confirmed the antioxidant activity of ST1-RGE in hydrogen peroxide-induced ROS production in HuH7 cells. ST1-RGE decreased ROS production in a dose-dependent manner, which suggested that C, EGCG and EC contributed to the inhibition of ROS production in hepatocytes. These results agree with the DPPH^•^ scavenging results. Accordingly, one study reported that green tea catechins blocked the increase of ROS production and DNA damage in carcinogen-induced human breast epithelial (MCF10A) cells [20]. Importantly, previous studies reported that green tea extract and EGCG removed intracellular labile iron pools (LIP) and ROS in iron-loaded mouse hepatocytes and human hepatoma (HepG2) cells, and lowered liver iron content in ferrocene-loaded β-knockout thalassemia mice [8,21]. We have recently determined that green tea extract rich in EGCG could reduce levels of ROS production in iron-loaded rat insulinoma pancreatic β-cells resulting in an improvement in pancreatic functions [22]. The resulting data support the determination that the detection of C, EGCG and EC in ST1-RGE was in accordance with their degree of antioxidant potency. These findings suggest that ST1-RGE contained C, EGCG and EC; therefore, ST1-RGE could prevent oxidative stress through its antioxidant activities.

Iron is an essential trace element that controls many cellular functions in the body. However, excessive iron can be deposited in cells and cause oxidative stress through the Fenton reaction, which can result in oxidative cellular damage and organ dysfunction. Consequently, iron chelators are widely used to decrease iron levels and improve cellular functions in iron-overloaded patients, such as thalassemia patients. At present, DFO and DFP are standard iron chelators for the treatment of β-thalassemia patients with iron overload. In this study, we examined the iron-binding and chelating activities of ST1-RGE. Similar to the reference hexadentate DFO and bidentate DFP chelators, ST1-RGE was bound with both ferrous and ferric ions in a concentration-dependent manner. Previous studies reported that green tea catechins, including EGCG, EC and C, exerted iron-binding/iron-chelating activities in biological systems involving renal tubules [23]. For instance, EGCG can bind with iron in a ratio of 2:1, while EC can bind with iron in a ratio of 3:1. In addition to the binding assay, we also evaluated the chelating activity of ST1-RGE on serum NTBI levels in thalassemic patients with iron overload. In the present study, DFO, DFP and ST1-RGE chelated serum NTBI levels in a dose-dependent manner. Although ST1-RGE was not as efficient as the standard chelators, DFO and DFP, it should be noted that ST1-RGE contained a combination of phenolic compounds which would vary in their chelating activities. In agreement with the iron-binding results, C, EGCG and EC in ST1-RGE could act as iron chelators that decreased the levels of serum NTBI in thalassemic patients. Likewise, our previous study revealed that isolated C and EGCG obtained from green tea chelated NTBI in serum thalassemic patients [24]. Moreover, another study found that green tea extract that was rich in EGCG reduced plasma malondialdehyde (MDA) and NTBI in β-knockout thalassemic mice with iron overload [25]. Thus, these results imply that ST1-RGE containing C, EGCG and EC could chelate iron and consequently reduce levels of serum NTBI.

The MTT test-based viability assay indicated that ST1-RGE was not toxic to both HuH7 and HepG2 cells at concentrations up to 200 μg/mL for 24 h. However, the treatment of ST1-RGE at 200 μg/mL for 48 h effectively reduced cell viability. It is possible that the high concentrations associated with this process, as well as the longer time of treatment (48 h), may have acted as conditions that were pro-oxidant, which could then generate ROS and cause toxicity to the cells. Previous studies have reported that EGCG at high doses could significantly induce ROS generation in primary rat hepatocytes, while EGCG at low doses could inhibit ROS formation. This outcome implies that EGCG showed toxicity to hepatocytes at high concentrations [26]. This finding is in agreement with our cytotoxicity results of the detected compounds in ST1-RGE. These results further suggest that not only the dose of ST1-RGE, but also the time of treatment would be associated with the degree of cytotoxicity of hepatocytes.

We further tested the acute toxicity of ST1-RGE in mice and found that a single dose of the drink did not result in any toxic signs or change the behavior of the mice during 14 days of observation. This result implies that ST1-RGE did not cause acute toxicity in mice with an LD_50_ value higher than 5 g/kg. Our result is in agreement with that of a previous study which reported that tested substances with LD_50_ values greater than 5000 mg/kg administered by oral route can be considered safe and/or nontoxic [27]. In addition to acute toxicity, we also confirmed the sub-chronic toxicity of ST1-RGE in Wistar rats, while most of hematological and biochemical parameters were not changed. However, administration of both doses of ST1-RGE for 90 d resulted in increased WBC numbers (but remained within a normal range) and increased ALT and AST activities (slightly higher than the normal range) when compared with the DI group [28]. However, there were no changes in liver weights and histopathological findings of the liver. These findings suggest that ST1-RGE could slightly alter hepatocytes but were not toxic to the liver.

## 4. Materials and Methods

### 4.1. Chemicals and Reagents

Acetic acid, ethyl acetate, absolute ethanol and methanol were purchased from BDH Chemicals Company, Poole, UK. Desferrioxamine mesylate (DFO or Desferal^®^) was purchased from a drug store in Maharaj Nakorn Chiang Mai Hospital, Faculty of Medicine, Chiang Mai University, Chiang Mai, Thailand. Catechin, DFP, 2′,7′-dichlorofluorescein diacetate reduced form (DCFH-DA), dimethylsulfoxide (DMSO), 2,2-diphenyl-1-picrylhydrazyl (DPPH), catechin (C), epicatechin (EC), epigallocatechin-3-gallate (EGCG), ferric ammonium citrate (FAC), ferric chloride, ferrous ammonium sulfate (FAS), fetal bovine serum (FBS), penicillin G, streptomycin, 6-hydroxy-2,5,7,8-tetramethylchroman-2-carboxylic acid (Trolox), thiobarbituric acid (TBA), 1,1,3,3-tetramethoxypropane (TMP), 3-(*N*-morpholino)propanesulfonic acid (MOPS), 3-(4,5-dimethylthiazol-2-yl)-2,5-diphenyltetrazolium bromide (MTT), metaphosphoric acid and nitrilotriacetic acid (NTA) were obtained from Sigma-Aldrich Chemicals Company (St. Louis, MO, USA). Dulbecco’s Modified Eagle Medium (DMEM), phosphate-buffered saline (PBS) and tryptic soy agar (TSA) were purchased from Invitrogen (GIBCO^™^, Thermo Fisher Scientific Corporation, Waltham, MA, USA). SOD assay kit was purchased from Sigma Chemical Company Limited, St. Louis, MO, USA.

### 4.2. Preparation of Sukhothai-1 Rice Grass Extract and Drink

Briefly, organic Sukhothai-1 rice grass (ST1-RG) lots 1, 2 and 3 aged 12–15 d were dried in a hot air oven at 50 °C for 2 h, extracted in boiling water (5.0 g/100 mL) at 80–98 °C for 6–9 min and filtered through filter paper (Whatman No. 1, cellulose type). The filtrate was placed in a glass bottle (40 mL capacity) that was attached with a label listing the major ingredients and served as an ST1-RG drink. The filtrate was then either frozen (−20 °C overnight) or dried in a vacuum microwave cabinet (2880 W, −40 to −80 °C, 2–3 millibar) and then served as both an ST1-RG extract (ST1-RGE) powder and a drink (260 mg/40 mL or 260 mg/bottle yield) [4]. Subsequently, ST1-RGE was kept in polypropylene bags (5 mg/bag), wrapped with aluminium foil sheets and kept under vacuum conditions in a freezer (−20 °C) until being used. Alternatively, the drink was stored in a refrigerator at 4 °C.

### 4.3. HPLC-DAD Quantification of Catechins

ST1-RGE was dissolved in 1% (*w*/*v*) DMSO, filtered on a membrane (polysulfone type, 0.45 μm pore size) and C, EC and EGCG were quantified using the high-performance liquid chromatography-diode array detector (HPLC-DAD) method established by Risner with slight modifications [29]. The HPLC system involved the use of the Agilent Technologies 1260 Infinity Quaternary System, an analytical column (C18 type, 150 mm × 4.6 mm, 5 μm particle size), linear-gradient elution with solvent A (3% acetic acid) and solvent B (100% methanol), a flow rate of 1.0 mL/min and photodiode array (PDA) detection at 280 nm. Authentic C, EC and EGCG (0–1000 μM each) were used to position the eluate from the column and to construct standard curves by plotting the concentrations on the *x*-axis against optical density (OD) at 280 nm on the *y*-axis.

### 4.4. Microbiological Test

Total colony numbers and minimal inhibitory concentration (MIC) values of pathogenic bacteria, including *Staphylococcus aureus* and *Escherichia coli*, were assayed in the Central Laboratory (Thailand). Company Limited, Chiang Mai Branch, Chiang Mai, Thailand using the modified agar-well diffusion method [30]. Aliquots of the ST1-RG drink were prepared at different concentrations (10–100 mg/mL). One hundred microliters of bacterial suspension containing 1 × 10^8^ colony-forming unit (CFU)/mL was streaked on a tryptic soy agar (TSA) plate. Discs were impregnated with the ST1-RG drink (12.5 µL), placed in the middle of the TSA agar inoculated with *E. coli* and *S. aureus* and incubated at 27 ± 2 °C for 48 h. Tests were carried out in triplicate, while a clear zone on the TSA was considered representative of the MIC value.

*Aspergilla flavus* toxins (Aflatoxin) B_1_, B_2_, G_1_ and G_2_ were analyzed in the Central Laboratory (Thailand) Company Limited, Chiang Mai Branch, Chiang Mai, Thailand using a multifunctional column (Mycosep) according to the AOAC Official Method 994.08 [31]. In the assay, the ST1-RG drink (250 mL) was extracted with the solvent (acetonitrile:H_2_O = 9:1, *v*/*v*) (100 mL) in a blender jar for 1 h and passed through filter paper (cellulose type, Whatman No. 1, Maidstone, UK). Stock solutions of the standard aflatoxin B_1_ (300 ng/mL), B_2_, (50 ng/mL), G_1_ (150 ng/mL) and G_2_ (50 ng/mL) were initially prepared in the solvent mixture (benzene:acetontrile = 98:2, *v*/*v*) and were then diluted ten times with acetonitrile before being used. The filtrate or standard Aflatoxin derivatives (3.0 mL) were applied to a multifunctional (reverse-phase, ion exclusion, ion exchange adsorbents) column (lipophilic charge active resin Mycosep™ 224 MFC, 10 mm × 100 mm, 6 mL capacity, Romer Labs, Inc., Washington, MO, USA) and eluted with mobile-phase solvent (acetonitrile–methanol–H_2_O = 1:1:4, *v*/*v/v*). Lipids, proteins, pigments and carbohydrates were adsorbed to the column resin, while aflatoxin derivatives were eluted from the column. The resulting eluates (0.2 mL) were incubated with the derivatizing solution (trifluoroacetic acid–glacial acetic acid–distilled water = 2:1:7, *v*/*v*/*v*) at 65 °C for 10 min and analyzed using a high-performance liquid chromatography fluorescence detector (HPLC-FLD) system as follows: injection volume (50 μL), column cartridge (C18 type, 4.6 mm × 100 mm, 5 μm particle size, Brownlee LC), a flow rate of 2.0 mL/min, FLD at an excitation wavelength 360 nm and emission wavelength 440 nm. Amounts of aflatoxin derivatives were calculated from a calibration curve constructed by plotting various concentrations of standard aflatoxin derivatives on the *x*-axis against fluorescence intensity (FI) on the *y*-axis.

### 4.5. Metal Analysis

Arsenic (As), copper (Cu), lead (Pb), mercury (Hg), tin (Sn) and zinc (Zn) in the ST1-RG drink samples (Code: CM59/03413-001 and Code CM59/03413-002) were analyzed in the Central Laboratory (Thailand) Company Limited, Chiang Mai Branch, Chiang Mai, Thailand using the inductively coupled plasma mass spectrometry (ICP-MS) technique according to the EPA 3052 Guidelines [32]. The low limit of detection of the ICP-MS analysis was 1 mg/kg.

### 4.6. Antioxidant Activity Assay

#### 4.6.1. DPPH^•^ Scavenging Assay

ST1-RGE (0–100 mg/mL), which had been dissolved in 1% DMSO or Trolox (0–0.25 mg/mL), were incubated with 0.4 mM DPPH^•^ solution in the dark for 30 min and OD was photometrically measured at 517 nm [33]. The ST1-RGE or Trolox doses were increased until decolorization of DPPH^•^ was complete. ROS scavenging activity was expressed as inhibition of DPPH^•^ production and calculated using the following equation:% Inhibition of DPPH^•^ production = [1 − OD_ST1-RG drink_/OD_DMSO_] × 100(1)

#### 4.6.2. Assay of Cellular ROS Scavenging Activity

Cellular ROS was probed with DCFH-DA fluorochrome and the resulting FI was measured using spectrofluorometry [34]. HepG2 cells were grown in DMEM supplemented with 10% FBS, penicillin (100 IU/mL) and streptomycin (100 IU/mL) in humidified 5% CO_2_ incubator at 37 °C and harvested at 80–90% confluence [35]. In the assay, the cells were pretreated with ST1-RGE (3.3, 10, 20 and 40 μg EGCG equivalent/mL) for 60 min, stimulated with 125 µM H_2_O_2_ solution (ROS generator) for 30 min and washed with PBS at a pH of 7.4. Finally, the cells were incubated with 20 μM DCFH-DA solution for 30 min in the dark and FI was measured using a spectrofluorometer (excitation wavelength 485 nm, emission wavelength 535 nm).

#### 4.6.3. Anti-Lipid Peroxidation Activity Assay

TBA reacted with aldehydes that were released from the lipid peroxidation of plasma lipoproteins and membrane phospholipids to yield a pink-colored TBARS, which could then be determined photometrically [25]. HepG2 cells (1.5 × 10^6^/well) were seeded onto a 6-well plate for 24 h and then loaded with 1 mM FAC for a further 24 h. The cells were incubated with 1% DMSO (control), DFP (50 μM equal to 9 μg/mL), ST1-RGE (5, 10 and 20 μg EGCG equivalent/mL) and combinations of the two for 24 h at 37 °C. The treated cells were harvested, washed twice with PBS and centrifuged at 300× *g* for 10 min. Cell pellets were then collected and resuspended in PBS. Cell lysates were prepared by placing cells in a Bioruptor water bath sonicator (conditions were set at 24 sec ON, 24 sec OFF) housed in a cold room (5–6 °C) for 2–3 min and centrifuged at 10,000× *g* for 15 min. Total protein content was determined using Bradford’s reagent [36]. Lipid-peroxidation products were measured photometrically using the TBARS assay. Cell lysate or standard MDA was mixed with 1% metaphosphoric acid and 0.67% TBA, and then incubated in a water bath at 95 °C for 60 min. After being cooled, *n*-butanol was added and the mixtures were centrifuged at 4500 rpm for 5 min. The OD of the *n*-butanol phase was measured at 535 nm using a double beam UV-VIS spectrophotometer. Standard curves were prepared over the concentration range 0.312–20 µM TMP.

#### 4.6.4. Determination of Serum SOD Activity in Rats

SOD is an important anti-oxidative enzyme that catalyzes a dismutation of superoxide anion (O_2_^•−^) into H_2_O_2_ and O_2_. An indirect method was used for determination of SOD activity using Dojindo’s highly water-soluble tetrazolium salt, 2-(4-iodophenyl)-3-(4-nitrophenyl)-5-(2,4-disulfophenyl)-2H-tetrazolium and monosodium salt (WST-1) as an electron acceptor that would then be reduced to a water-soluble WST-1 formazan product exhibiting an absorption spectrum at 440 nm [37]. Wistar rats (5 male and 5 female each) were divided into four groups and orally administered with DI and ST1-RGE (50 and 100 mg/kg BW/d) for 90 days. Blood was collected from tail veins on days 0, 45 and 90, and serum was separated for analysis of SOD activity using a commercial SOD assay kit according to the manufacturer’s instructions. In the analysis, serum (20 μL) was mixed with WST working solution (200 μL), incubated with xanthine/xanthine oxidase working solution at 37 °C for 20 min and OD was measured at 440 nm using a 96-well microplate reader. SOD activity was calculated using the appropriate equation, for which the OD of WST-1 formazan was directly proportional to the amount of O_2_^•−^. A standard curve of SOD was prepared using the standard provided in the kit, and the SOD activity value for each sample was read from this curve. SOD activity was expressed as U/mL reagent, for which one unit represents the amount of SOD that inhibited the rate of formazan dye formation by 50%.

### 4.7. Assay of Iron-Chelating Activity

Ferric nitrilotriacetate (Fe-NTA) was prepared by incubating 1 mM ferric chloride with 10 mM NTA at a pH of 7.0 for 1 h. In the assay, ST1-RGE (0–20 mg/mL), DFO (0–200 μmol/L) and DFP (0–200 μmol/L) were incubated with FAC (500 μmol/L) or Fe-NTA (500 μmol/L) at room temperature for 60 min. OD values were then measured at 390 nm to produce the iron-ST1-RGE chelate, 430 nm to produce the iron-DFO chelate and 450 nm to produce the iron-DFP chelate using a double-beam spectrophotometer [4,24].

Additionally, the chelating activity of NTBI was determined using the iron-sensitive fluorescent bead method established by Ma and colleagues [38]. Serum was incubated with 800 mM NTA, pH 7.0 solution at room temperature for 30 min and centrifuged through a filtration membrane (10-kDa cut off, polysulfone type) at 10,000 rpm, 15 °C for 50 min to obtain the Fe-NTA filtrate. The filtrate and standard iron solution were then incubated with an iron-sensitive probe (2.8 × 10^6^ beads in 50 mM MOPS buffer) pH 7.4 at 37 °C for 16 h, and the FI was measured with a flow cytometer (Guava^®^ easyCyte™ 6HT-2L, Luminex Corporation, Merck KGaA, Darmstadt, Germany). The system was set up at an excitation wavelength of 495 nm and an emission wavelength of 525 nm. Guava InCyte software was used for data acquisition and analysis. The population gates were selected from dot-plots of the control bead area and FI of 5000 events. For the purposes of calibration, beads were mixed with normal human serum as a control and were used to set up the FI at 100%, while the relative FI of the beads with patient serum was calculated accordingly. The Fe calibration curve was established between peak height values against the standard iron solution (0–10 μM ferric-NTA in 80 mM NTA solution). Nonlinear regression assuming variable slope sigmoidal dose–response function was used to determine NTBI concentrations.

### 4.8. Toxicity Study

#### 4.8.1. Cytotoxicity

Cytotoxicity was determined using the MTT method [39]. Human hepatocellular carcinoma (HepG2 and HuH7) cells were cultured in 10% FBS in DMEM supplemented with penicillin G (100 U/mL) and streptomycin (100 μg/mL) at 37 °C in the humidified atmosphere of a 5% CO_2_ incubator. Different concentrations of ST1-RGE (0–200 μg/mL) were prepared in 1% DMSO. Cultured hepatocytes were treated with ST1-RGE at 37 °C for 24 and 48 h and incubated with MTT reagent at 37 °C for 4 h. Produced formazan was dissolved with 1% DMSO and OD was measured at wavelengths of 540 nm and 630 nm against the reagent blank, for which any differences in OD were recorded between the two wavelengths. Viable cells were determined using the following formula:% Viable cells = 1 − (OD_DMSO_ − OD_ST1-RGE_)/OD_DMSO_ × 100(2)
where OD_DMSO_ denotes OD from the cells without treatment and OD_ST1-RG_ drink denotes OD from the cells treated with ST1-RGE. A curve of cell survival was established by plotting the OD values (*y*-axis) against ST1-RGE concentrations (*x*-axis). A concentration at 80% inhibition of cell viability (IC_80_) was then determined from the graph.

#### 4.8.2. Acute Toxicity in Mice

Acute oral toxicity of ST1-RGE (Sample number 195852131003) was assessed in mice at the Institute of Medicinal Plant Research (Certified ISO/IEC 17025: 2005), Department of Medical Science, Ministry of Public Health, Nonthaburi, Thailand according to the Organization for Economic Co-operation and Development (OECD) Test Guidelines 423, 2001. Mice (Balc, *n* = 3) were fasted for 3–4 h prior to being weighed and given the specified doses. Brown lyophilized ST1-RGE (0.5 g/mL) was prepared in DI and orally administered to the mice (10 mL/kg BW) using a gavage needle. Mice were observed after dosing at least once during the first 30 min, periodically during the first 24 h and then daily for a total of 14 d. All observations, including signs of toxicity, changes in skin and fur, eyes and mucous membranes, vital systems and behavior were recorded systematically with individual records being maintained for each animal. At the end of the study period, surviving mice were enumerated and a median lethal dose at 50% (LD_50_) of the sample was determined. All study mice, including those that were dead or removed during the test, were subject to gross necropsy for the appropriate records. Histopathology of internal organs and tissues was investigated in case evidence of any gross pathology were to become evident in the findings.

#### 4.8.3. Sub-Chronic Oral Toxicity in Rats

Sub-chronic toxicity test was performed according to the OECD Test Guidelines [40]. Wistar rats (5 male and 5 female) were divided into four groups and orally administered with DI and ST1-RGE (50 and 100 mg/kg BW/d) for 90 d. A recovery (satellite) group was established following the main dosing phase and extended the study for an additional 28 d to understand whether toxicities observed at the end of the dosing phase were partially or completely reversible, and then to confirm the persistence or reversibility of the resulting effects. Satellite groups of DI and ST1-RGE were included in nonclinical toxicity studies. Body weights were recorded weekly, while blood samples were collected from tail veins on days 0 and 45 for further analysis. At the end of the study, rats were sacrificed using phenobarbital anesthesia and heart blood samples were collected from the left ventricle for parameter analysis. Internal organs were then dissected, weighed and prepared for pathological examination.

For the laboratory investigation, CBC was determined using an automated cell analyzer (Beckman Coulter, Brea, CA, USA) according to the manufacturer’s instructions. Concentrations of BUN, CRE, TG, TC, HDL-C and LDL-C, and activities of ALT, AST and ALP in the serum were measured using an automated Clinical Chemistry analyzer (Randox^®^ Laboratories, County Antrim, UK) according to the manufacturer’s instructions. For the pathological examination, organs were observed and fixed in 10% neutral buffered formaldehyde for microscopic investigation. Tissues were dissected, trimmed (4 mm thick), placed in appropriately labeled cassettes and dehydrated in increasing concentrations of ethanol. The alcohol was then removed with xylene and specimens were embedded in the paraffin block, cut into extremely thin tissue sections (around 5 μm thick) with a microtome, stained with hematoxylin and eosin dyes, histopathologically examined in five non-overlapping fields under a light microscope (Nikon ECLIPSE E200, Nikon Corporation, Tokyo, Japan) and photographed by a clinical pathologist. The image processing method was used for analysis of the histology sections, while digital images were analyzed using ImageJ Ver.1.47i software.

### 4.9. Statistical Analysis

Data were analyzed using SPSS program (IBM version 18.0) and presented as mean ± SD or mean ± SEM values. Statistical significance was determined using one-way analysis of variance (ANOVA). *p* < 0.05 was considered a significant difference.

## 5. Conclusions

This study demonstrated that the SukhoThai-1 rice grass extracts and the drinks containing catechins, particularly EGCG, showed potent radical-scavenging, anti-lipid peroxidation and iron-chelating properties in aqueous and biological systems. Moreover, the ST1-RG drink did not cause toxicity in hepatocytes, or acute and sub-chronic toxicity in animals. Overall, the findings suggest that the rice grass drink is safe for consumption and could potentially alleviate oxidative stress in thalassemia patients with iron overload, as well as cancer patients undergoing chemotherapy and who exhibit metabolic abnormalities.

## Figures and Tables

**Figure 1 molecules-26-06887-f001:**
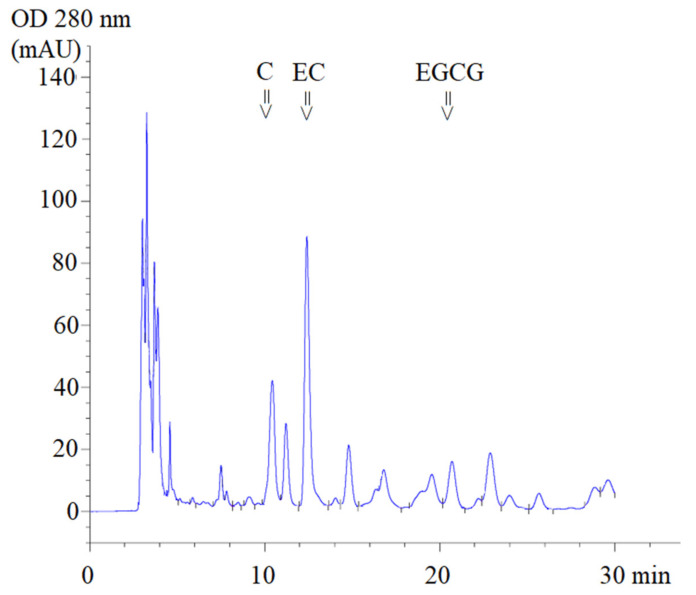
HPLC-DAD profile of Sukhothai-1 rice grass extract. C = catechin, EC = epicatechin, EGCG = epigallocatechin-3-gallate (EGCG), mAU = milli-absorbance unit and OD = optical density.

**Figure 2 molecules-26-06887-f002:**
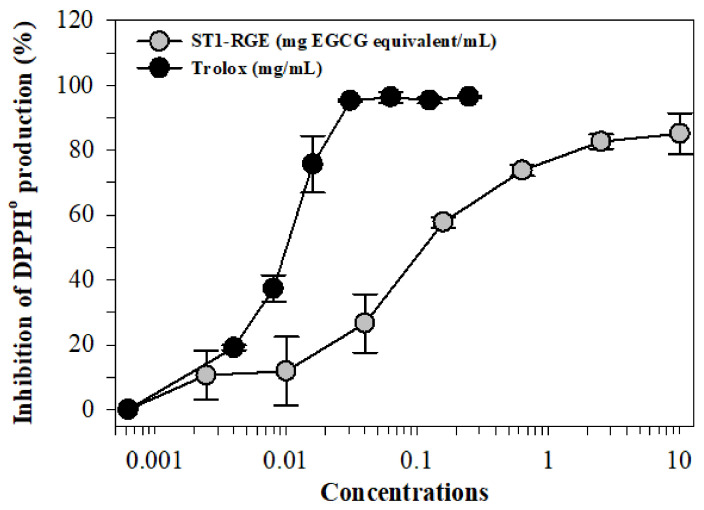
Inhibition of DPPH^•^ production by ST1-RGE and Trolox. Data obtained from three repetitive experiments are shown as mean ± SD values. Abbreviations: EGCG = epigallocatechin-3-gallate, DPPH = 2,2-diphenyl-1-picrylhydrazyl, ST1-RGE = Sukhothai-1 rice grass extract.

**Figure 3 molecules-26-06887-f003:**
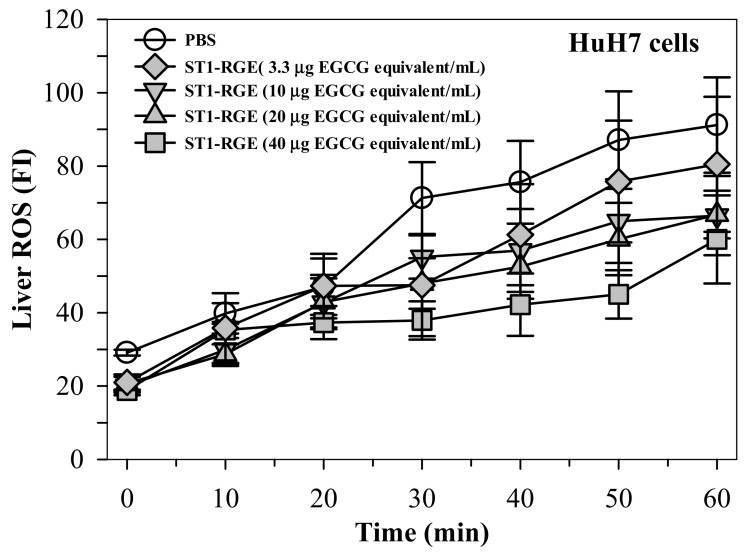
Radical-scavenging activity of ST1-RGE in liver cells. Data obtained from three repetitive experiments are shown as mean ± standard error values of the mean (SEM). Abbreviations: EGCG = epigallocatechin-3-gallate, FI = fluorescence intensity, PBS = phosphate-buffered saline, ROS = reactive oxygen species, ST1-RGE = Sukhothai-1 rice grass extract.

**Figure 4 molecules-26-06887-f004:**
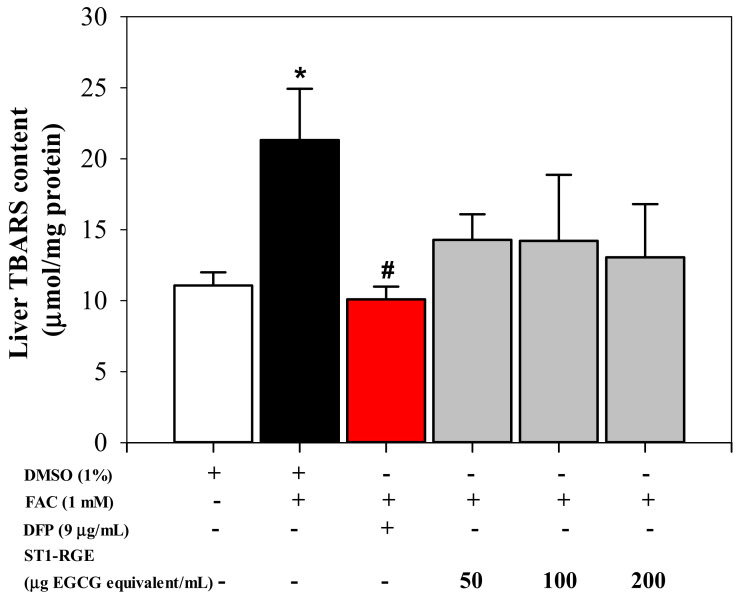
Anti-lipid peroxidation activity of ST1-RGE in liver cells. Data obtained from three repetitive experiments are shown as mean ± SEM values. * *p* < 0.05 when compared with DMSO; # *p* < when compared with DMSO + FAC. Abbreviations: DFP = deferiprone, DMSO = domethylsulfoxide, EGCG = epigallocatechin-3-gallate, FAC = ferric ammonium citrate, ST1-RGE = Sukhothai-1 rice grass extract, TBARS = thiobarbituric acid-reactive substances.

**Figure 5 molecules-26-06887-f005:**
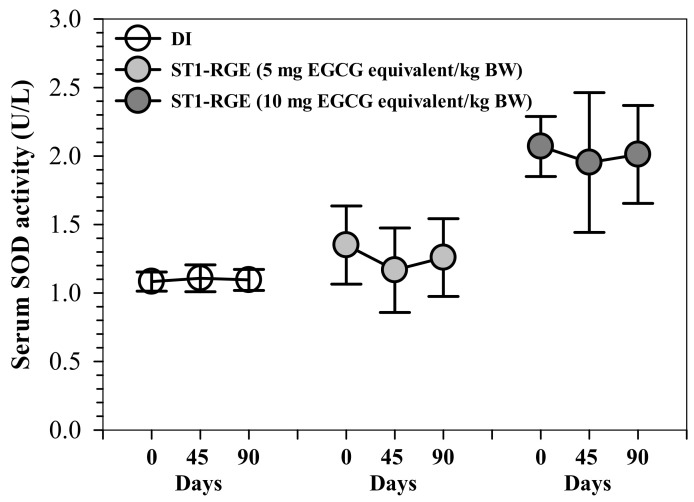
Serum SOD activity from rats that had been orally administered with DI and ST1-RGE. Data are shown as mean ± SD values (*n* = 6). Abbreviations: DI = deionized water, EGCG = epigallocatechin-3-gallate, SOD = superoxide dismutase, ST1-RGE = Sukhothai-1 rice grass extract.

**Figure 6 molecules-26-06887-f006:**
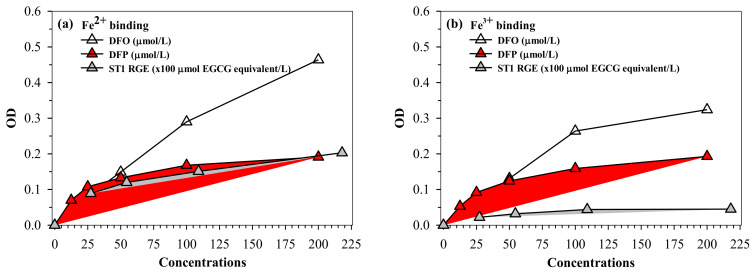
(**a**–**c**) Chelating activity of iron from FAS, FAC and NTBI by ST1-RGE, DFO and DFP. Data obtained from three repetitive experiments are expressed as mean ± SD values. Abbreviations: DFO = desferrioxamine, DFP = deferiprone, EGCG = epigallocatechin-3-gallate, FAC = ferric ammonium citrate, FAS = ferrous ammonium sulfate, Fe^2+^ = ferrous ion, Fe^3+^ = ferric ion, NTBI = non-transferrin bound iron, OD = optical density, ST1-RGE = Sukhothai-1 rice grass extract.

**Figure 7 molecules-26-06887-f007:**
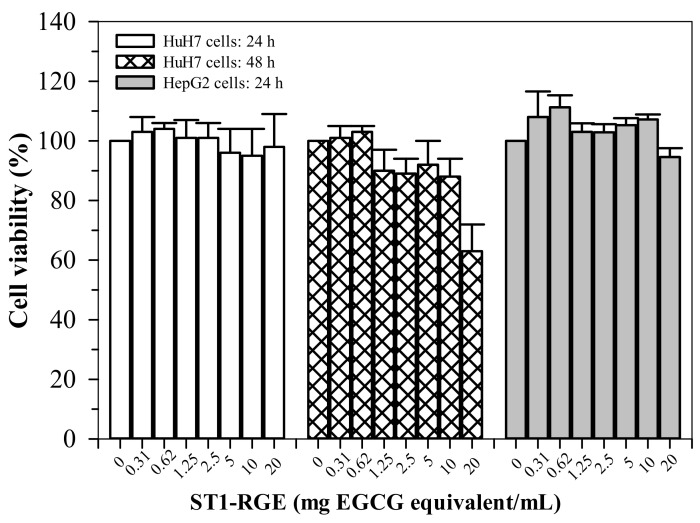
Viability of HuH7 and HepG2 cells treated with ST1-RGE. Data obtained from four repetitive experiments are expressed as mean ± standard error values of the mean (SEM). Abbreviations: EGCG = epigllocatechin-3-gallate, ST1-RGE = Sukhothai-1 rice grass extract.

**Figure 8 molecules-26-06887-f008:**
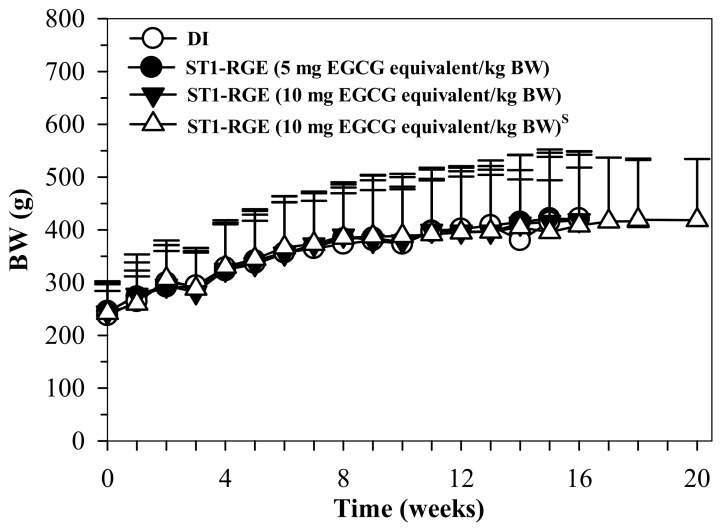
BW values of rats orally administered with DI and ST1-RGE for 90 g. Data are shown as mean ± SD values. Abbreviations: BW = body weight, DI = deionized water, EGCG = epigallocatechin-3-gallate, ST1-RGE = Sukhothai-1 rice grass extract. ^S^ (S superscript) indicates a satellite study.

**Table 1 molecules-26-06887-t001:** Amounts of C, EC and EGCG in ST1-RG extracts and drinks.

Sample	Extract (mg/g)			Drink (mg/40 mL Serving)		
	C	EGCG	EC	C	EGCG	EC
ST1-RG (lot 1)	13.7 ± 5.1	102.9 ± 4.6	7.1 ± 1.1	3.57	26.77	1.85
ST1-RG (lot 2)	10.4 ± 4.9	98.0 ± 4.5	3.8 ± 0.9 *	2.71	25.47	0.98
ST1-RG (lot 3)	11.5± 0.2	106.0 ± 6.8	4.4 ± 1.1 *	3.01	27.55	1.13

Data obtained from two separate analyses are shown as mean ± SD values. * *p* < 0.05 when compared with ST1-RG (lot 1). Abbreviations: C = catechin, EC = epicatechin, EGCG = epigallacatechin-3-gallate, ST1-RG = Sukhothai-1 rice grass.

**Table 2 molecules-26-06887-t002:** Amounts of elements found in the ST1-RG extract (**a**) and drink (**b**). Data obtained from two separate analyses are shown as mean ± SD values.

(a) ST1-RGE	As (mg/kg)	Cu (mg/kg)	Pb (mg/kg)	Hg (mg/kg)	Sn (mg/kg)	Zn (mg/kg)
ST1-RGE (lot 2)	4.26	18.26	0.98	ND	3.15	75.93
ST1-RGE (lot 3)	3.80	16.66	0.27	ND	5.81	61.90
Mean ± SD	4.03 ± 0.33	17.46 ± 1.13	0.63 ± 0.50	ND	4.48 ± 1.88	68.92 ± 9.92
**(b) ST1-RG Drink**	**As (μg/Serving)**	**Cu (μg/Serving)**	**Pb (μg/Serving)**	**Hg (μg/Serving)**	**Sn (μg/Serving)**	**Zn (μg/Serving)**
ST1 RG (1/1)	1.11	4.75	0.25	ND	0.82	19.74
ST1 RG (1/2)	0.99	4.33	0.07	ND	1.51	16.09
Mean ± SD	1.05 ± 0.08	4.53 ± 0.30	0.16 ± 0.13	ND	1.16 ± 0.40	17.92 ± 2.58

Data obtained from two separate analyses are shown as mean ± SD values. Abbreviations: As = arsenic, Cu = copper, Hg = mercury, Pb = lead, Sn = tin and Zn = zinc. ND = not detectable, ST1-RG = Sukhothai-1 rice grass.

**Table 3 molecules-26-06887-t003:** Levels of hematological parameters of rats treated with DI and ST1-RGE for 90 days.

(a) WBC Indices	DI		ST1-RGE (5 mg EGCG Equivalent/kg BW)		ST1-RGE (10 mg EGCG Equivalent/kg BW)		ST1-RGE (10 mg EGCG Equivalent/kg BW) ^S^	
	Day 0	Days 90	Day 0	Days 90	Day 0	Days 90	Day 0	Days 90
WBC number (×10^3^/μL)	1.95 ± 1.11	2.93 ± 1.72	5.89 ± 1.68	1.79 ± 8.3	6.12 ± 2.61	2.69 ± 1.39	3.74 ± 1.67	1.85 ± 0.88
Neutrophil (%)	18.8 ± 1.3	10.1 ± 4.6	11.1 ± 2.7	17.4 ± 2.8	13.7 ± 2.8	8.8 ± 4.4	13.7 ± 2.9	18.0 ± 3.9
Lymphocyte (%)	70.8 ± 5.9	79.8 ± 7.2	84.8 ± 3.0	73.4 ± 6.4	81.9 ± 3.4	81.2 ± 8.5	80.4 ± 3.8	69.1 ± 3.4
Monocyte (%)	8.1 ± 4.6	7.5 ± 3.9	3.0 ± 1.8	7.6 ± 5.9	2.2 ± 1.5	8.3 ± 5.7	4.7 ± 2.5	11.1 ± 9.0
Eosinophil (%)	2.3 ± 0.6	1.3 ± 0.8	1.2 ± 0.5	1.6 ± 0.6	2.1 ± 1.6	1.6 ± 0.6	1.1 ± 0.3	1.8 ± 1.2
Basophil (%)	0	0	0	0	0.1 ± 0.1	0	0	0
**(b) RBC Indices**	**DI**		**ST1-RGE** **(5 mg EGCG Equivalent/kg BW)**		**ST1-RGE** **(10 mg EGCG Equivalent/kg BW)**		**ST1-RGE** **(10 mg EGCG Equivalent/kg BW) ^S^**	
	**Day 0**	**Days 30**	**Day 0**	**Days 30**	**Day 0**	**Days 30**	**Day 0**	**Days 30**
RBC number (×10^6^/μL)	7.85 ± 0.96	8.15 ± 0.51	7.82 ± 0.38	7.28 ± 0.54	7.73 ± 1.10	7.69 ± 7.3	8.76 ± 0.39	6.79 ± 1.04
Hb (g/dL)	15.1 ± 1.7	15.8 ± 0.5	15.2 ± 0.5	14.8 ± 0.7	14.2 ± 1.6	15.5 ± 1.1	16.3 ± 0.9	14.1 ± 2.0
Hct (%)	46.5 ± 5.7	49.1 ± 2.4	46.0 ± 1.8	43.2 ± 3.0	45.0 ± 5.8	46.2 ± 4.4	51.6 ± 3.0	41.6 ± 6.5
MCV (fL)	59.2 ± 1.4	60.3 ± 1.4	58.8 ± 1.1	59.4 ± 0.9	58.3 ± 1.1	60.1 ± 0.9	58.9 ± 2.1	61.3 ± 1.0
MCH (pg)	19.2 ± 0.5	19.4 ± 0.8	19.5 ± 0.6	20.4 ± 0.6	19.0 ± 0.5	20.3 ± 0.7	18.6 ± 0.6	20.8 ± 0.5
MCHC (g/dL)	32.5 ± 0.5	32.2 ± 0.9	33.1 ± 0.7	34.3 ± 0.9	32.6 ± 0.7	33.7 ± 1.1	31.6 ± 0.4	34.1 ± 0.9
RDW (%)	12.1 ± 1.1	12.1 ± 0.7	11.8 ± 0.8	11.9 ± 0.6	11.5 ± 0.5	12.3 ± 0.4	11.4 ± 0.7
RDW (fL)	29.5 ± 2.9	31.5 ± 1.5	28.9 ± 2.2	28.9 ± 2.2	28.8 ± 1.2	28.4 ± 1.2	29.8 ± 1.0	29.1 ± 1.7
**(c) PLT Indices**	**DI**		**ST1-RGE(5 mg EGCG Equivalent/kg BW)**		**ST1-RGE** **(10 mg EGCG Equivalent/kg BW)**		**ST1-RGE** **(10 mg EGCG Equivalent/kg BW) ^S^**	
	**Day 0**	**Days 30**	**Day 0**	**Days 30**	**Day 0**	**Days 30**	**Day 0**	**Days 30**
PLT (×10^3^/μL)	664 ± 319	791 ± 73	612 ± 219	792 ± 229	333 ± 351	962 ± 812	511 ± 284	554 ± 351
MPV (fL)	6.2 ± 0.3	6.2 ± 0.3	6.1 ± 0.5	5.5 ± 0.1	6.6 ± 0.4	5.9 ± 0.2	6.4 ± 0.6	5.8 ± 0.2
PDW	15.3 ± 0.6	15.3 ± 0.6	15.1 ± 0.2	15.1 ± 0.2	15.6 ± 0.9	15.2 ± 0.1	15.1 ± 0.3	15.3 ± 0.2
Pct (%)	0.41 ± 0.20	0.41 + 0.20	0.37 ± 0.12	0.43 ± 0.12	0.21 ± 0.23	0.36 ± 0.14	0.31 ± 0.16	0.37 ± 0.19

Data are shown as mean ± SD values (3M, 3F). Abbreviations: BW = body weight, DI = deionized water, EGCG = epigallocatechin-3-gallate, Hb = hemoglobin, Hct = hematocrit, MCH = mean corpuscular hemoglobin, MCHC = mean corpuscular hemoglobin concentration, MPV = mean platelet volume, Pct = plateletocrit, PDW = platelet distribution width, PLT = platelet, RBC = red blood cells, RDW = red cell distribution width, ST1-RGE = Sukhothai-1-rice grass extract, WBC = white blood cells. ^S^ (S superscript) indicates a satellite study.

**Table 4 molecules-26-06887-t004:** Levels of serum biomarkers of rats treated with DI and ST1-RGE for 90 days.

Parameters	DI		ST1-RGE (50 mg EGCG Equivalent/kg BW)		ST1-RGE (100 mg EGCG Equivalent/kg BW)		ST1-RGE (100 mg EGCG Equivalent/kg BW) ^S^	
	Day 0	Days 90	Day 0	Days 90	Day 0	Days 90	Day 0	Days 90
BUN (mg.dL)	26.6 ± 5.1	25.4 ± 3.6	23.8 ± 6.2	22.4 ± 2.3	25.6 ± 4.2	22.2 ± 2.3	22.9 ± 1.8	23.9 ± 2.6
CRE (mg/dL)	0.8 ± 0.1	0.6 ± 0.1	0.8 ± 0.1	0.5 ± 0.2 *	0.7 ± 0.0	0.6 ± 0.1	0.7 ± 0.1	0.5 ± 0.1 *
AST activity (U/L)	127.3 ± 11.5	241.7 ± 43.8	110.9 ± 11.2	295.5 ± 89.3 *	111.6 ± 18.3	231.7 ± 63.8	126.4 ± 25.1	336.2 ± 173.6 *
ALT activity (U/L)	41.8 ± 15.2	66.1 ± 16.3	32.0 ± 7.0	72.8 ± 35.2 *	32.4 ± 5.9	65.0 ± 11.1	31.2 ± 7.3	86.4 ± 28.3 *
ALP activity (U/L)	110.6 ± 28.4	61.4 ± 9.8 *	105.8 ± 27.1	59.4 ± 12.5 *	110.3 ± 19.3	66.7 ± 8.2 *	114.7 ± 21.1	72.8 ± 14.5 *
Alb (g/dL)	3.8 ± 0.1	3.5 ± 0.2	3.8 ± 0.2	3.3 ± 0.2 *	3.8 ± 0.1	3.4 ± 0.3	3.7 ± 0.1	3.5 ± 0.5
TB (mg/dL)	0.76 ± 0.11	0.59 ± 0.10	0.79 ± 0.11	0.51 ± 0.16	0.73 ± 0.05	0.60 ± 0.11	0.72 ± 0.11	0.51 ± 0.10
TC (mg/dL)	100.5 ± 10.9	95.2 ± 12.7	95.0 ± 8.8	97.2 ± 6.1	93.5 ± 7.1	98.5 ± 9.5	98.0 ± 6.4	105.7 ± 19.1
TG (mg/dL)	86.3 ± 41.4	63.6 ± 11.5	75.5 ± 24.0	50.8 ± 12.9	86.5 ± 14.0	54.2 ± 18.8	69.5 ± 11.0	59.5 ± 13.4
HDL-C (mg/dL)	27.7 ± 3.1	24.9 ± 5.3	26.3 ± 2.4	25.1 ± 3.2	24.9 ± 2.6	26.7 ± 3.6	25.5 ± 2.7	39.9 ± 5.6
LDL-C (mg/dL)	8.5 ± 1.9	5.9 ± 1.9	7.9 ± 2.5	12.5 ± 2.5 *	9.1 ± 3.1	12.2 ± 3.2	7.3 ± 2.1	7.1 ± 1.7

Data are shown as mean ± SD values (3M, 3F). *p* < 0.05 when compared with day 0. * *p* < 0.05 when compared with D0. Abbreviations: Alb = albumin, ALT = alanine aminotransferase, ALP = alkaline phosphatase, AST = aspartate aminotransferase, BUN = blood urea nitrogen, BW = body weight, CRE = creatinine, DI = deionized water, EGCG = epigallocatechin-3-gallate, HDLC = high-density lipoprotein cholesterol, LDLC = low-density lipoprotein cholesterol, NA = not available, ST1-RGE = Sukhothai-1-rice grass extract, TB = total bilirubin, TC = total cholesterol, TG = triglyceride. ^S^ (S superscript) indicates a satellite study.

**Table 5 molecules-26-06887-t005:** Weight of organs taken from rats treated with DI and ST1-RGE for 90 days.

Organs	Organ Wet Weight (g)			
	DI	ST1-RGE (50 mg EGCG Equivalent/kg BW)	ST1-RGE (100 mg EGCG Equivalent/kg BW)	ST1-RGE (100 mg EGCG Equivalent/kg BW) ^S^
Heart	1.36 ± 0.38	1.44 ± 0.41	1.36 ± 0.30	1.33 ± 0.24
Liver	10.60 ± 3.84	11.3 ± 4.86	10.63 ± 3.48	12.36 ± 2.86
Lungs	2.79 ± 3.38	1.67 ± 0.28	1.57 ± 0.32	1.50 ± 0.20
Spleen	0.78 ± 0.14	0.88 ± 0.23	0.77 ± 0.20	0.87 ± 0.11
Left kidney	1.30 ± 0.47	1.28 ± 0.67	1.36 ± 0.45	1.36 ± 0.33
Right kidney	1.22 ± 0.40	1.42 ± 0.44	1.28 ± 0.39	1.38 ± 0.36
Kidneys	2.52 ± 0.83	2.7 ± 1.08 *	2.63 ± 0.83 *	2.74 ± 0.69 *
Left adrenal gland	0.04 ± 0.01	0.04 ± 0.01	0.03 ± 0.01	0.03 ± 0.01
Right adrenal gland	0.04 ± 0.01	0.04 ± 0.01	0.04 ± 0.00	0.03 ± 0.01
Adrenal glands	0.07 ± 0.01	0.07 ± 0.02 *	0.07 ± 0.01 *	0.06 ± 0.01 *
Brain	2.09 ± 0.12	2.12 ± 0.11	2.05 ± 0.09	2.08 ± 0.09
Left testis	2.10 ± 0.06	0.12 ± 0.02	0.15 ± 0.05	0.14 ± 0.01
Right testis	2.09 ± 0.10	1.44 ± 0.41	1.36 ± 0.30	1.33 ± 0.24
Testes	4.19 ± 0.15	1.67 ± 0.28	1.57 ± 0.32	1.50 ± 0.20
Left ovary	0.04 ± 0.01	1.28 ± 0.67	1.36 ± 0.45	1.36 ± 0.33
Right ovary	0.05 ± 0.02	1.42 ± 0.44	1.28 ± 0.39	1.38 ± 0.36
Ovaries	0.09 ± 0.01	2.70 ± 1.08	2.63 ± 0.83	2.74 ± 0.69

Data are shown as mean ± SD values (3M, 3F). * *p* < 0.05 when compared with D0. Abbreviation: BW = body weight, DI = deionized water, EGCG = epigallocatechin-3-gallate, F = female, M = male, ST1-RGE = Sukhothai1-rice grass extract. ^S^ (S superscript) indicates a satellite study.

## Data Availability

The study did not report any data.

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
