# Peer review of "Identifying Chemical Composition, Safety and Bioactivity of Thai Rice Grass Extract Drink in Cells and Animals"

_molecules, 2021, doi:10.3390/molecules26226887_

Round 1

Reviewer 1 Report

Identifying Chemical Composition, Safety and Bioactivity of Thai Rice-Grass Extract Drink in Cells and Animals

Journal: Molecules

The manuscript is interesting and well within the scope of the journal.

However, there are some aspects that have to be clarified.

Abstract

….revealed numerous phenolic compounds… How many compounds were identified?

…limits when compared with provisional guideline values. With respect to what ?

EGCG ….? Define the first time it is used.

  1. Introduction

Update references.

Figure 1. Delete:“Extracto de pasto de arroz Sukhothai-1 (ST1-RGE) analizado usando el método HPLC-DAD como se describe en la Sección 4.3, para el cual se colocaron compuestos auténticos de catequina (C), epicatequina (EC) y epigalocatequina-3-galato (EGCG) en la TR de 10,4, 12,4 y 20,5 min, respectively.” That is part of the methodology.

Table 1. Delete: "Sukhothai-1 rice grass (ST1-RG) extracts and drinks were analyzed using HPLC-DAD (as has been described in Section 4.3) and positioned by authentic C (10.4 min), EC (12.4 min) and EGCG (20.5 min). Concentrations of C, EC and EGCG were determined from calibration curves of the authentic standards."

"Data obtained from two separate analyzes are shown as mean + SD values." Write in the footer of the Table. Statistical analysis of the data is missing.

Table 2. Delete “Sukhothai-1 rice grass (ST1-RG) extract (a) and drink (b) were digested in concentrated nitric acid and hydrofluoric acid using microwave heating. They were then analyzed using the ICP-MS method as has been described in Section 4.5.”

“Data obtained from two separate analyses are shown as mean+SD values.” Write in the footer of the Table. The standard deviation is missing. Statistical analysis of the data is missing.

Figure 2. Delete “DPPH radical solution was incubated with various concentrations of ST1-RGE or Trolox, while OD was photometrically measured at 517 nm.” X-axis units are missing.

Figure 3. Delete “HuH7 cells were challenged with H2O2, treated with PBS, while various concentrations of ST1-RGE and ROS was detected with DCFH-DA fluorochrome using flow cytometry.”

Figure 4. Delete “HuH7 cells were loaded with 1 mM FAC, then treated with 1% DMSO, 9 mg/mL DFP, or 50, 100 and 200 mg EGCG-equivalent ST1-RGE/mL. Lipid-peroxidation products were colorimetrically measured using the TBARS method.” That is part of the methodology.

Figure 5. Delete “Wistar rats were orally administered with deionized water (DI) and Sukhothai 1 rice grass extract (ST1-RGE) for 90 d and superoxide dismutase (SOD) activity was determined in the serum on days 0, 45 and 90.” That is part of the methodology.

Figure 6. X-axis units are missing.

Figure 7. Delete “Human hepatocellular carcinoma (HuH7 and HepG2) cells were incubated with various concentrations of Sukhothai-1 rice grass extract (ST1-RGE) for 24 and 48 h, while their viability was determned using the MTT method. That is part of the methodology.

Figure 8. Delete “Wistar rats were orally administered with deion- ized water (DI) and Sukhothai 1 rice grass extract (ST1-RGE) for 90 d in the standard study and for 118 d in the satellite study. BWs were then recorded weekly.” That is part of the methodology.

Table 3. Delete “Wistar rats were orally administered with DI and ST1-RGE for 90 d in the standard study and an additional 28 d in the satellite study (S). Blood samples were collected on day 0 and days 90 or 118 of the study, while complete blood count parameters were determined.” That is part of the methodology. Statistical analysis of the data is missing.

“Data are shown as mean+SD values (3M, 3F).” Write in the footer of the Table.

Table 4. Delete “Wistar rats were orally administered with DI and ST1-RGE for 90 d in the standard study and an additional 28 d in the satellite study (S). Blood samples were collected on days 0 and 90 or 118 of the study and serum levels of biochemical markers were determined.” That is part of the methodology.

“Data are shown as mean+SD values (3M, 3F).” Write in the footer of the Table. Statistical analysis of the data is missing.

Table 5. Delete “Wistar rats were orally administered with DI and ST1-RGE for 90 d in the standard study and an additional 28 d in the satellite study (S). The rats were sacrified and internal organs were collected on days 90 or 118 of the study for pathologcal examination.” That is part of the methodology.

 “Data are shown as mean+SD values (3M, 3F).” Write in the footer of the Table. Statistical analysis of the data is missing.

The manuscript is interesting, however, apparently it was not ready for submission to the journal, one of the fundamental parts in writing a research article is the statistical analysis of the data. 

Reviewer 2 Report

This manucript talked about the rice grass extract drink, did a lot of work, which is interesting and valuable. Some suggestions: 

  1.  The meaning for ST1-RG (1/2-2), ST1-RG (1/1) and ST1-RG (1/2)?
  2. Fig 1 should be edited, remove pink line in fig.
  3. many figs and tables did not mark the significant difference btween different groups, eg: Fig 5, Fig7 and Table3....

Round 2

Reviewer 1 Report

The authors tended the observations made to the manuscript.